# Aberrant computational mechanisms of social learning and decision-making in schizophrenia and borderline personality disorder

Lara Henco[1,2]*, Andreea O. Diaconescu[3,4,5], Juha M. Lahnakoski[1,6,7], Marie-Luise Brandi[1], Sophia Hörmann[1], Johannes Hennings[8], Alkomiet Hasan[9,10], Irina Papazova[9], Wolfgang Strube[9], Dimitris Bolis[1,11], Leonhard Schilbach[1,2,11,12☉], Christoph Mathys[4,13,14☉]

1 Independent Max Planck Research Group for Social Neuroscience, Max Planck Institute of Psychiatry, Munich, Germany, 2 Graduate School for Systemic Neurosciences, Munich, Germany, 3 Department of Psychiatry (UPK), University of Basel, Basel, Switzerland, 4 Translational Neuromodeling Unit (TNU), Institute for Biomedical Engineering, University of Zurich and ETH Zurich, Zurich, Switzerland, 5 Krembil Centre for Neuroinformatics, Centre for Addiction and Mental Health (CAMH), University of Toronto, Canada, 6 Institute of Neuroscience and Medicine, Brain & Behaviour (INM-7), Research Centre Jülich, Jülich, Germany, 7 Institute of Systems Neuroscience, Medical Faculty, Heinrich Heine University Düsseldorf, Düsseldorf, Germany, 8 Department of Dialectical Behavioral Therapy, kbo-Isar-Amper-Klinikum Munich-East, Munich/Haar, Germany, 9 Department of Psychiatry and Psychotherapy, University Hospital Munich, LMU Munich, Munich, Germany, 10 Department of Psychiatry, Psychotherapy and Psychosomatic, University of Augsburg, Medical Faculty, Augsburg, Germany, 11 International Max Planck Research School for Translational Psychiatry (IMPRS-TP), Munich, Germany, 12 Medical Faculty, LMU Munich, Munich, Germany, 13 Scuola Internazionale Superiore di Studi Avanzati (SISSA),Trieste, Italy, 14 Interacting Minds Centre, Aarhus University, Aarhus, Denmark

☉ These authors contributed equally to this work.
* lara_henco@psych.mpg.de

**Data Availability Statement:** Applicable German federal law does not allow public archiving or peer-to-peer sharing of individual raw data in this case.

## Abstract

Psychiatric disorders are ubiquitously characterized by debilitating social impairments. These difficulties are thought to emerge from aberrant social inference. In order to elucidate the underlying computational mechanisms, patients diagnosed with major depressive disorder (N = 29), schizophrenia (N = 31), and borderline personality disorder (N = 31) as well as healthy controls (N = 34) performed a probabilistic reward learning task in which participants could learn from social and non-social information. Patients with schizophrenia and borderline personality disorder performed more poorly on the task than healthy controls and patients with major depressive disorder. Broken down by domain, borderline personality disorder patients performed better in the social compared to the non-social domain. In contrast, controls and major depressive disorder patients showed the opposite pattern and schizophrenia patients showed no difference between domains. In effect, borderline personality disorder patients gave up a possible overall performance advantage by concentrating their learning in the social at the expense of the non-social domain. We used computational modeling to assess learning and decision-making parameters estimated for each participant from their behavior. This enabled additional insights into the underlying learning and decision-making mechanisms. Patients with borderline personality disorder showed slower

However, the processed data underlying the main results of the study, together with the computational modeling code and behavioral analyses are available here: https://osf.io/8kfph/.

**Funding:** NO - The funders had no role in study design, data collection and analysis, decision to publish, or preparation of the manuscript.

**Competing interests:** The authors have declared that no competing interests exist.

learning from social and non-social information and an exaggerated sensitivity to changes in environmental volatility, both in the non-social and the social domain, but more so in the latter. Regarding decision-making the modeling revealed that compared to controls and major depression patients, patients with borderline personality disorder and schizophrenia showed a stronger reliance on social relative to non-social information when making choices. Depressed patients did not differ significantly from controls in this respect. Overall, our results are consistent with the notion of a general interpersonal hypersensitivity in borderline personality disorder and schizophrenia based on a shared computational mechanism characterized by an over-reliance on beliefs about others in making decisions and by an exaggerated need to make sense of others during learning specifically in borderline personality disorder.

## Author summary

People suffering from psychiatric disorders frequently experience difficulties in social interaction, such as an impaired ability to use social signals to build representations of others and use these to guide behavior. Computational models of learning and decision-making enable the characterization of individual patterns in learning and decision-making mechanisms that may be disorder-specific or disorder-general. We employed this approach to investigate the behavior of healthy participants and patients diagnosed with depression, schizophrenia, and borderline personality disorder while they performed a probabilistic reward learning task which included a social component. Patients with schizophrenia and borderline personality disorder performed more poorly on the task than controls and depressed patients. In addition, patients with borderline personality disorder concentrated their learning efforts more on the social compared to the non-social information. Computational modeling additionally revealed that borderline personality disorder patients showed a reduced flexibility in the weighting of newly obtained social and non-social information when learning about their predictive value. Instead, we found exaggerated learning of the volatility of social and non-social information. Additionally, we found a pattern shared between patients with borderline personality disorder and schizophrenia who both showed an over-reliance on predictions about social information during decision-making. Our modeling therefore provides a computational account of the exaggerated need to make sense of and rely on one's interpretation of others' behavior, which is prominent in both disorders.

## Introduction

Impairments in social cognition are frequently experienced by people suffering from a psychiatric disorder. For instance, patients with major depressive disorder (MDD) and schizophrenia (SCZ) show a reduction in (social) reward sensitivity and motivation to engage in social interactions [1–5]. Despite high levels of social anhedonia, patients with SCZ show a tendency to over-interpret the meaning of social signals [6]. Individuals with borderline personality disorder (BPD) suffer from rapidly changing beliefs about others that polarise between approach and rejection [7]. Together, these impairments are associated with aberrant inferences/beliefs about oneself and the social environment.

In computational terms, the emergence of aberrant inference can be ascribed to an impaired ability to adjust learning in response to environmental changes [8]. Bayesian learning

models allow for a parsimonious algorithmic description of changes in beliefs relevant for accurate inference: belief updates can be written as a surprise signal (prediction error) weighted by a learning rate [9]. The learning rate depends on the ratio between the precision of the sensory data and the precision of the prior belief [10,11]. Whereas healthy participants increase their learning rate more strongly in volatile compared to stable environments [12,13], patients with autism do so less owing to an over-estimation of environmental volatility [8]. Impairments in the estimation of environmental volatility have also been studied as a mechanism for psychosis and SCZ [14–17] as well as MDD [18]. One recent study found that, unlike healthy controls, participants with BPD did not show an increase in learning when social and reward contingencies became volatile [19]. The authors suggested that this might be due to higher expected baseline volatility in participants with BPD. However, the computational model employed in that study did not explicitly model beliefs about volatility.

Adopting previous suggestions of aberrant volatility learning in psychiatric disorders and its role in impaired probability learning, the current study employed Bayesian hierarchical modeling to investigate probabilistic social inference in a volatile context across three major psychiatric disorders, which have previously been associated with social impairments: MDD, SCZ and BPD. Here, the current study investigated whether volatility and probability learning is equally affected when inferring on the hidden states of non-social and social outcomes across the three different disorders. We further asked whether aberrant social learning and decision-making were associated with differences in social anhedonia.

To this end, we adopted a probabilistic reward learning task (introduced in [20]), in which participants could learn from two types of information: non-social and social information. In order to probe the spontaneous rather than explicitly instructed use of social information as in previous social learning studies [12,13,21,22], we did not explicitly tell participants to learn about the social information. We used the hierarchical Gaussian filter (HGF; [10,11]) to obtain a profile of each participant's particular way of updating beliefs when receiving social and non-social information while making decisions in a volatile context. The HGF is a generic hierarchical Bayesian inference model for volatile environments with parameters that reflect individual variations in cognitive style. We went beyond other recent computational psychiatry studies using the HGF (e.g., [8,23–27]) in that we used two parallel HGF hierarchies for social and non-social aspects of the environment (cf. [20,29]). Our modeling framework was specifically designed also to quantify the relative weight participants afforded their beliefs about the predictive value of social compared to non-social information in decision-making.

## Materials and methods

### Ethics statement

All participants were naïve to the purpose of the experiment and provided written informed consent to take part in the study after a written and verbal explanation of the study procedure. The study was in line with the Declaration of Helsinki and approval for the experimental protocol was granted by the local ethics committee of the Medical Faculty of the Ludwig-Maximilians University of Munich.

### Participants

Patients were recruited for the present study after an independent and experienced clinician diagnosed them using ICD-10 criteria for 1) a depressive episode (F32), schizophrenia (F20.0) and emotionally unstable personality disorder (F60.3). HC and patients with MDD were recruited through the Max Planck Institute of Psychiatry. Patients with SCZ were recruited at the Department of Psychiatry and Psychotherapy at the University Hospital Munich. Patients

with BPD were recruited at the kbo-Isar-Amper-Klinikum in Haar, Munich. Participants were chosen prior to analysis such that groups were matched for age ($\chi^2$ = 5.302, $P$ = 0.151; Kruskal-Wallis one-way non-parametric ANOVA because of difference in age variance between groups, see S1 Table). Exclusion criteria were a history of neurological disease or injury, reported substance abuse at the time of the investigation, a history of electroconvulsive therapy, and diagnoses of comorbid personality disorder in the case of MDD and SCZ. Furthermore, 9 participants had to be excluded from the analysis due to one of the following reasons: unsaved data due to technical problems (1 HC, 2 BPD). Prior participation in another study which involved the same paradigm (1 HC), always picking the card with the higher reward value (1 HC), either following (1 SCZ) or going against (1 BPD) the gaze on more than 95% of trials (indicating a learning-free strategy), interruption of the task (1 SCZ), change to the diagnosis following study participation (1 MDD). The final sample consisted of 31 HC, 28 MDD, 29 SCZ and 28 BPD. We additionally acquired psychometric data (S1 Table) to further characterize the participants: All patients were asked to fill out questionnaires measuring autistic traits with the autism spectrum quotient (AQ [30]) and social anhedonia symptoms with the Anticipatory and Consummatory Interpersonal Pleasure Scale (ACIPS; [31]). We additionally assessed positive and negative symptoms using the Positive and Negative Syndrome Scale (PANSS [32]) and mood symptoms using the Calgary Depression Scale for Schizophrenia (CDSS [33]) in patients with SCZ. To assess the severity of Borderline Personality Disorder we used the short version of the Borderline Symptom List (BSL-23 [34]). Additional questionnaires were employed but analyzed within the scope of a different study and therefore not presented here. Demographic data as well as details regarding the medication can be seen in S2 Table.

## Experimental paradigm and procedure

After giving informed consent, participants were seated in front of a computer screen in a quiet room where they received the task instructions. In the same probabilistic learning task as in [20,29], participants were asked to choose between one of two cards (blue or green) in order to maximize their score which was converted into a monetary reward (1–6 €) that was added to participants' compensation at the end of the task. An animated face was displayed between the cards, which first gazed down, then up towards the participant, before it shifted its gaze towards one of the cards (Fig 1A). The blue and green card appeared randomly on the left and right side from the face and participants responded using 'a' or 'l' on a German QWERTZ keyboard. When a response was logged within the allowed time (6000 ms), the chosen card was marked for 1000 ms until the outcome (correct: green check mark/wrong: red cross) was displayed for 1000 ms. When the correct card was chosen, the reward value (1–9) displayed on the card was added to the score. Participants were instructed that these values were not associated with the cards' winning probabilities, but that they might want to choose the card with the higher value if they were completely uncertain about the outcome. When the wrong card was chosen or participants failed to choose a card in the allotted time, the score remained unchanged. Participants were told that the cards had winning probabilities that changed in the course of the experiment but they were not informed about the systematic association between the face animation's gaze and the trial outcome. Specifically, they were not told that the probability with which the face animation pointed towards the winning card on a given trial varied systematically throughout the task according to the schedule given in Fig 1B. Instead, we simply told participants that the face was integrated into the task to make it more interesting. The probabilistic schedules for social and non-social information were independent from each other in order to estimate participant-specific learning rates separately for both types of information. In the first half of the experiment (trials 1–60), the card winning probabilities were

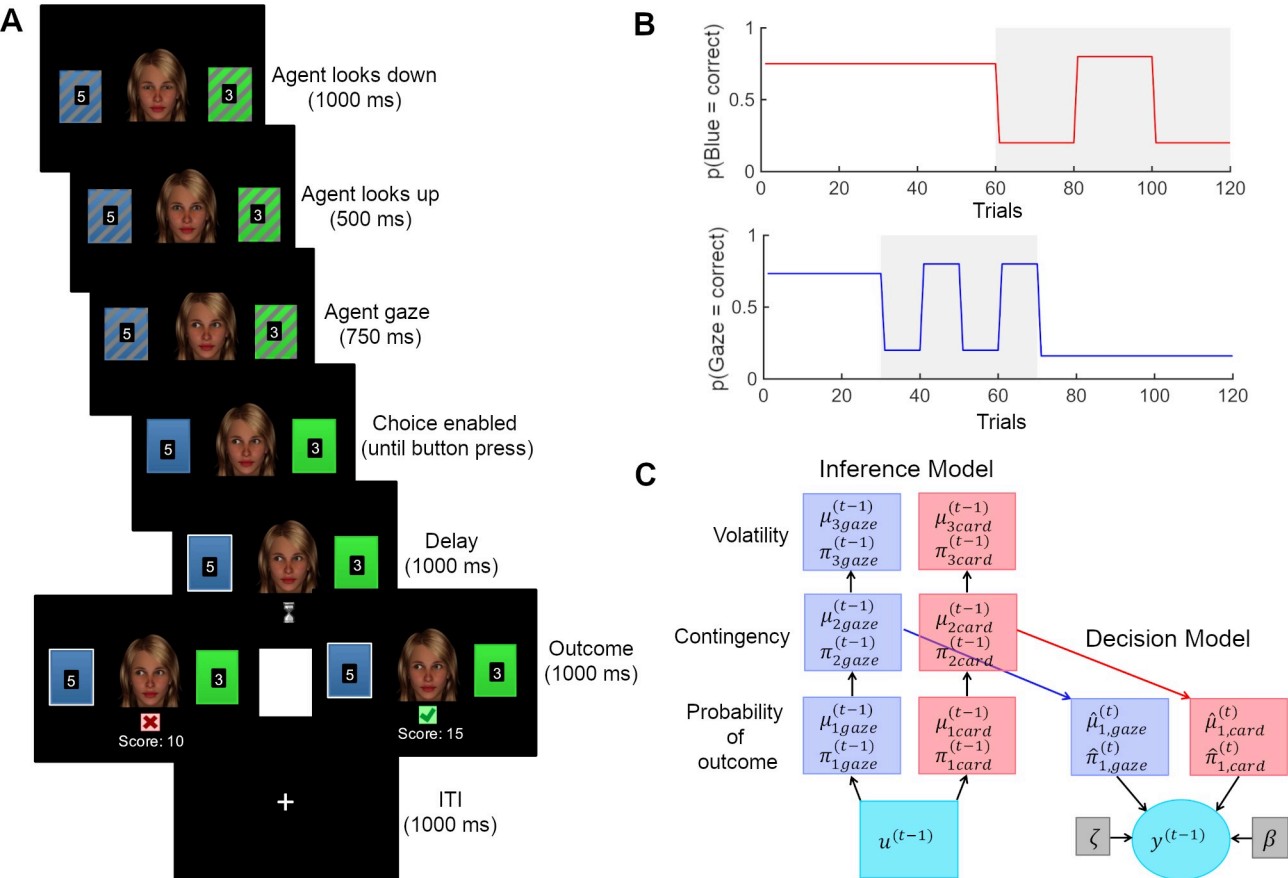

**Fig 1. Task design and computational decision and inference model.** (A) Participants were asked to make a choice between blue and green cards after the gray shading on the colored rectangles (cards) had disappeared (i.e., 750 ms after the face shifted its gaze towards one of the cards). After a delay phase, the outcome was presented (correct/wrong). If the choice was correct, the reward amount (number on the chosen card) was added to a cumulative score. The task consisted of 120 trials. (B) Probability schedules from which outcomes were drawn. Volatile phases are marked in grey. (C) Posteriors are deterministic functions of predictions and outcomes. Predictions in turn are deterministic functions of the posteriors of previous trials. Decisions $y^{(t)}$ are probabilistically determined by predictions and decision model parameters $\zeta$ and $\beta$. Deterministic quantities are presented as boxes and probabilistic quantities in circles.

stable, whereas in the second half (trials 61–120) they changed (volatile phase). The social cue had a stable contingency during trials 1–30 and trials 71–120, whereas contingency was volatile during trials 31–70. We used two types of schedules for the social cue which were each presented to half of the participants. In one schedule (depicted in Fig 1B), the probability of the social cue looking towards the winning card was 73% in the first stable phase (trial 1–30) and therefore started as congruent to the winning card (congruent-first). The second probability schedule was flipped, so that the probability of the social cue looking towards the winning card was 27% in the first stable phase (incongruent-first). In total, 15 control participants received the congruent-first schedule, 15 participants with MDD, 14 with SCZ and 15 with BPD. Positions of the cards on the screen (blue left or right) were determined randomly. The task was programmed and presented with PsyToolkit [35].

## Computational modeling

**Observing the observer.** We modeled behavior in the 'observing the observer' (OTO) framework [36,37]. This entails a *response model*, which probabilistically predicts a

participant's choices based on their inferred beliefs, and a *perceptual model*, on which the response model depends because it describes the trajectories of participants' inferred beliefs based on experimental inputs. The OTO framework is conceptually very similar to the idea of *inverse reinforcement learning* [38].

**Perceptual models.** We used three different perceptual models in order to make inferences on the most likely mechanisms of learning in our paradigm. We used a Bayesian HGF as well as two non-Bayesian learning models, the Sutton K1 model [39] and a Rescorla Wagner model [40]. All three were implemented in the HGF toolbox, such that they perform parallel learning about the social (predictive value of gaze) and non-social (predictive value of card color) aspects of the task environment.

While the Rescorla Wagner learning model assumes fixed domain-specific learning rates for learning the social and non-social information, the Sutton K1 model assumes variable learning rates that are scaled by recent prediction errors. In contrast, the HGF takes into accout that beliefs have different degrees of uncertainty, scaling the learning rate dynamically as a function of uncertainty. In addition, the HGF assumes hierchical learning of different aspects of the environment. On the lowest level of the hierarchy, agents learn about concrete events (i.e. stimuli), whereas at higher levels of the hierarchy, agents learn about more abstract features of the environment, such as probabilistic associations between stimuli and how these change in time (i.e. volatility). Learning at every level is driven by a ratio between the precision of the input (from the level below) and the precision of prior beliefs.

The HGF is an inference model resulting from the inversion of a generative model in which states of the world are coupled in a three-level hierarchy: At the lowest level of the generative model, $x_{1_{gaze}}$ and $x_{1_{card}}$ represent the two inputs in a binary form (social cue: 1 = correct, 0 = incorrect; card outcome: 1 = blue wins, 0 = green wins). Level $x_{2_{gaze}}$ and $x_{2_{card}}$ represent the tendency of the gaze to be correct and the tendency of the blue card to win. State $x_{2_{gaze}}$ and $x_{2_{card}}$ evolve as first-order autoregressive (AR(1)) processes with a step size determined by the state at the third level. Level $x_{3_{gaze}}$ and $x_{3_{card}}$ represent the log-volatility of the two tendencies and also evolve as first-order autoregressive (AR(1)) processes. The probabilities of $x_{1_{gaze}} = 1$ and $x_{1_{card}} = 1$ are the logistic sigmoid transformations of $x_{2_{gaze}}$ and $x_{2_{card}}$ (Eq 1).

$$p\left(x_1^{(t)} = 1\right) = \frac{1}{1 + \exp(-x_2^{(t)})} \tag{1}$$

Participants' responses $y$ were coded with respect to the congruency with the 'advice' (1 = follow; 0 = not follow) and were used to invert the model in order to infer the belief trajectories at all three levels $i$ = 1,2,3.

On every trial $k$, the beliefs $\mu_i^{(k)}$ (and their precisions $\pi_i^{(k)}$) about the environmental states at the $i$-th level are updated via prediction errors $\delta_{i-1}^{(k)}$ from the level below weighted by a precision ratio $\psi_i^{(k)}$ (Eqs 2 and 3). This means that belief updates are larger (due to higher precision weights) when the precision of the posterior belief ($\pi_2^{(k)}$ or $\pi_3^{(k)}$) is low and the precision of the prediction $\hat{\pi}_2^{(k)}$ is high. Consequently, prediction errors are weighted more during phases of high volatility (cf. S1 Fig, panel C, dotted blue trajectory). For the analysis, we used $q(\psi_2^{(k)})$ (Eq 4), which is a transformation of $\psi_2^{(k)}$ (Eq 3) (cf. [41], supplementary material) that corrects for the sigmoid mapping between first and second level, effectively making $q(\psi_2^{(k)})$ an uncertainty

(inverse precision) measure for first-level beliefs.

$$\Delta\mu_i^{(k)} \; \propto \; \psi_i^{(k)}\delta_{i-1}^{(k)} \; (i = 2, 3) \tag{2}$$

$$\psi_2^{(k)} = \frac{1}{\pi_2^{(k)}} \tag{3}$$

$$q(\psi_2^{(k)}) = \psi_2^{(k)} \, s(\mu_2^{(k)})(1 - s(\mu_2^{(k)})) \tag{4}$$

$$\psi_3^{(k)} = \frac{\hat{\pi}_2^{(k)}}{\pi_3^{(k)}} \tag{5}$$

Participant-specific parameters $\omega_{2card}$ and $\omega_{2gaze}$ represent the learning rates at the second level, i.e. the speed at which association strengths change. Correspondingly, $\omega_{3card}$ and $\omega_{3gaze}$ represent the learning rates of the volatilities.

**Response models.**   In the response model a combined belief $b^{(t)}$ (Eq 6) was mapped onto decisions, which resulted from a combination of both the inferred prediction $\hat{\mu}_{1,gaze}^{(t)}$ that the face animation's gaze will go to the winning card and the inferred prediction $\hat{\mu}_{1,card}^{(t)}$ that the color of the card that the gaze went to would win (see example in S1 Fig). The inferred prediction $\hat{\mu}_{1,gaze}^{(t)}$ and $\hat{\mu}_{1,card}^{(t)}$ were weighted by $w_{gaze}^{(t)}$ and $w_{card}^{(t)}$ (Eqs 7 and 8), which are functions of the respective precisions ($\hat{\pi}_{1,gaze}^{(t)}$ and $\hat{\pi}_{1,card}^{(t)}$, Eqs 9 and 10). The precisions (Eqs 9 and 10) represent the inverse variances of a Bernoulli distribution of $\hat{\mu}_{1,gaze}^{(t)}$ and $\hat{\mu}_{1,card}^{(t)}$.

The constant parameter $\zeta$ represents the weight on the precision of the social prediction compared to the precision of the non-social prediction (Eq 7). In other words, this parameter describes the propensity to weight the social over the non-social information. We investigated the effect of varying the social weighting factor $\zeta$, by simulating the combined belief $b^{(t)}$ (Eq 6) of agents with same perceptual parameters (fixed at prior values as depicted in S3 Table) but different $\zeta$ values.

$$b^{(t)} = w_{gaze}^{(t)} \, \hat{\mu}_{1,gaze}^{(t)} + w_{card}^{(t)} \, \hat{\mu}_{1,card}^{(t)} \tag{6}$$

$$w_{gaze}^{(t)} = \frac{\zeta\hat{\pi}_{1,gaze}^{(t)}}{\zeta\hat{\pi}_{1,gaze}^{(t)} + \hat{\pi}_{1,card}^{(t)}} \tag{7}$$

$$w_{card}^{(t)} = \frac{\hat{\pi}_{1,card}^{(t)}}{\zeta\hat{\pi}_{1,gaze}^{(t)} + \hat{\pi}_{1,card}^{(t)}} \tag{8}$$

$$\hat{\pi}_{1,gaze}^{(t)} = \frac{1}{\hat{\mu}_{1,gaze}^{(t)}(1 - \hat{\mu}_{1,gaze}^{(t)})} \tag{9}$$

$$\hat{\pi}_{1,card}^{(t)} = \frac{1}{\hat{\mu}_{1,card}^{(t)}(1 - \hat{\mu}_{1,card}^{(t)})} \tag{10}$$

In the response model, we used the combined belief $b^{(t)}$ (Eq 6) in a logistic sigmoid (softmax) function to model the probability $Prob_{gaze}^{(t)}$ (Eq 11). In this function, the belief was weighted by the predicted reward of the card when the advice is taken $r_{gaze}$ or not $r_{notgaze}$

(Eq 11). We took account of possible subject-specific non-linear distortions in weighting the expected reward by using a weighted average (parameter $\eta$) of linear and logarithmic weighting of expected reward.

$$
\begin{aligned}
prob_{gaze} = p(y^{(t)} = 1) = \eta/(1 + \exp(-\gamma^{(t)}(r_{gaze}^{(t)} b^{(t)} - r_{notgaze}^{(t)}(1 - b^{(t)}))))+ \\
(1 - \eta)/(1 + \exp(-\gamma^{(t)}(\log(r_{gaze}^{(t)}) b^{(t)} - \log(r_{notgaze}^{(t)})(1 - b^{(t)}))))
\end{aligned} \tag{11}
$$

The mapping of beliefs onto actions varied as a function of the inverse decision temperature $\gamma^{(t)}$, where large $\gamma^{(t)}$ implied a high alignment between belief and choice (low decision noise) and a smaller $\gamma^{(t)}$ a low alignment between belief and choice (high decision noise). Our four different response models varied in terms of how $\gamma^{(t)}$ was defined. In response model 1, $\gamma^{(t)}$ was a combination of the log-volatility of the third level for both cues combined with constant participant-specific decision noise $\beta$ (Eq 12). In response model 2, $\gamma^{(t)}$ was a combination of the log-volatility of the third level for the social cue and participant-specific decision noise (Eq 13) and in response model 3, $\gamma^{(t)}$ was a combination of the log-volatility of the third level for the non-social cue and participant-specific decision noise (Eq 14). In model 4, $\gamma^{(t)}$ only included the participant-specific decision noise (Eq 15).

$$
1)\ \gamma^{(t)} = \beta \exp(-\hat{\mu}_{3,card}^{(t)} - \hat{\mu}_{3,gaze}^{(t)}) \tag{12}
$$

$$
2)\ \gamma^{(t)} = \beta \exp(-\hat{\mu}_{3,gaze}^{(t)}) \tag{13}
$$

$$
3)\ \gamma^{(t)} = \beta \exp(-\hat{\mu}_{3,card}^{(t)}) \tag{14}
$$

$$
4)\ \gamma^{(t)} = \beta \tag{15}
$$

We used the HGF toolbox, version 4.1, which is part of the software package TAPAS (https://translationalneuromodeling.github.io/tapas) for parameter estimation. We fitted six alternate combinations of perceptual and response models, which were subjected to random-effects Bayesian Model Selection [42,43] (spm_BMS in SPM12; http://www.fil.ion.ucl.uk/spm). The HGF was combined with all four response models. The non-hierarchical models were combined with response model 4 only, owing to the lack of third-level belief trajectories. Details of the prior settings of all models can be seen in S3 Table.

We additionally included two non-learning models, which assume that participants repeat the actions that lead to reward and switch the strategy immediately after loss (Win-Stay-Lose-Shift) and a model that assumes random responding throughout the task (random-responding). These models were adapted from [44] and implemented in the HGF toolbox to calculate the log model evidence.

**Model comparison and validity.** The log model evidence (LME) for each participant and each model were subjected to Bayesian Model Selection [43,44] (spm_BMS in SPM12). This procedure estimates the expected posterior probabilities (EXP_P), i.e. the posterior probability of the prevalence of each model in the population, the exceedance probability (XP), i.e. the probability that a given model outperforms all others in the comparison, and the more conservative protected exceedance probability (PXP), which additionally considers the possibility that all models are equally good. We additionally performed within-participant model comparisons to identify the model with the highest LME for each participant (cf. S4 Table). This turned up 16 subjects where random responding had a higher model evidence than the overall winning model. Excluding these participants did not change the pattern of results (cf. S7–S9 Tables).

**Posterior predictive validity of model parameters and Parameter Recovery.** To test the adequacy of the model, we simulated responses based on the estimated parameters from the winning model for each participant 10 times, resulting in 1160 simulations. As for the real data, we then calculated percentage of high reward probability choices for the two cues and the two phases, subjecting them to the same ANOVA that was performed with the real behavioral data. In this way, we checked whether the simulated responses produce the same group differences in response accuracy. In a second step, simulated responses were again used to invert the winning model to check whether model parameters could be recovered. For each subject, we calculated the average parameter values estimated from the simulated data and correlated (Pearson's correlations) them with the values estimated from the real data, which served as input for the simulation.

**Regression-based choice sequence analysis.** We ran a regression based choice sequence analysis in order to investigate adaptation to environmental volatility without needing to fit a learning model. According to the HGF, agents should increase their learning rate in more volatile environments, giving more weight to recent outcomes. In more stable environments, agents should adopt lower learning rates, affording less weight to recent outcomes in order to better filter out noise. To test this in a model-agnostic way, we implemented two general linear models (GLMs) with the responses $y$ as outcome variables for the 'card' and 'gaze' space respectively (card: 1 = blue taken; 0 = green taken; gaze: 1 = follow; 0 = not follow). For each GLM, we included as predictors the outcomes of the past 5 trials (t-1:t-5: 1 = blue correct; 0 = green correct) and (t-1:t-5: 1 = gaze correct; 0 = gaze incorrect) as well as two predictors for the expected reward (1–9) for either card winning. We ran these GLMs for the stable and volatile phases separately. The slopes of the coefficient estimates of the past outcome predictors were taken as a model-agnostic readout of the 'learning rate', indiciating the degree to which more recent information is weighted. A t-test was applied to compare the difference between these slopes during stable and volatile phases of the task.

**Statistical analysis.** Performance (% correct responses) was subjected to a one-way ANOVA with group (HC vs. MDD vs. SCZ vs. BPD) and schedule (congruent first vs. incongruent first) as between-subject factors.

In order to understand task performance in a domain-specific way, we additionally calculated response accuracy based on the ground truth reward probability (high probability choices) of both cues (non-social and social) and both phases (stable and volatile). The proportion of high probability choices was subjected to a mixed ANOVA with Cue Type (Non-Social vs. Social) and Phase (Stable vs. Volatile) as within-subject factors and Group (HC vs. MDD vs. SCZ vs. BPD) and schedule (congruent first vs. incongruent first) as between-subject factors. Advice taking (advice followed or not on a given trial) was subjected to a mixed ANOVA with social accuracy (high vs. low) and schedule stability (stable vs. volatile) as within-subject factors. Group (HC vs. MDD vs. SCZ vs. BPD) and schedule (congruent first vs. incongruent first) were included as between-subject factors.

Mean precision weights on the second and third level ($q(\psi_2)$ and $\psi_3$) separately entered two mixed ANOVAs as dependent variables with schedule stability as a within-subject factor (stable vs.volatile), information type as within participants factor (social vs. non-social). The group (HC vs. MDD vs. SCZ vs. BPD) and schedule (congruent first vs. incongruent first) were between subject factors.

We subjected the posterior estimate for $\zeta$ to a one-way ANOVA with group (HC vs. MDD vs. SCZ vs. BPD) as between-subject factor and schedule (congruent first vs. incongruent first) as a covariate.

We hypothesized that social anhedonia (measured by the Anticipatory and Consummatory Interpersonal Pleasure Scale, ACIPS) would be associated with a reduction in learning in the

social domain. To test this, we first performed a one-way ANOVA with ACIPS scores as dependent variable and group as the factor (HC vs. MDD vs. SCZ vs. BPD) followed by a multivariate regression with ACIPS as dependent variable and the social learning rates $\omega_{2gaze}$ and the weighting factor $\zeta$ as predictors of social learning and decision making. The group factor (HC vs. MDD vs. SCZ vs. BPD) was entered as covariate. This analysis was done for all participants who completed the ACIPS questionnaire (n = 106 of $n_{total}$ = 116).

All ANOVA post hoc $t$ tests were Bonferroni-corrected for multiple comparisons. All $p$-values are two-tailed with a significance threshold of $p < .05$. Statistical tests were performed using JASP (Version 0.9 2.0; https://jasp-stats.org/) or Matlab (Version 2018b; https://mathworks.com).

## Results

### Behavior

There was a significant difference between the groups in the overall performance, i.e. % of rewarded responses ($F(3,108)$ = 7.504, $p<0.001$, $\eta^2$ = 0.167): Post-hoc comparisons showed that both patients with SCZ and BPD performed significantly worse compared to HC and patients with MDD (SCZ–HC $t$ = 3.781, $p_{bonf}$ = 0.002, $d$ = 0.994, SCZ–MDD $t$ = 2.817, $p_{bonf}$ = 0.035, $d$ = 0.745, BPD–HC $t$ = 3.732, $p_{bonf}$ = 0.002, $d$ = 0.979, BPD–MDD $t$ = 2.78, $p_{bonf}$ = 0.038, $d$ = 0.732). There was no significant difference in performance between patients with BPD and SCZ ($t$ = -0.01, $p_{bonf}$ = 1.000, $d$ = -0.003) nor between HC and patients with MDD ($t$ = 0.88, $p_{bonf}$ = 1.000, $d$ = 0.240). Performance was not significantly affected by the schedule order (congruent first vs. incongruent first; $F(1,108)$ = 0.027, $p$ = 0.870, $\eta^2$ = 0) or its interaction with the patient groups ($F(3,108)$ = 1.302, $p$ = 0.278, $\eta^2$ = 0.029).

There was a significant difference between the groups in the overall proportion of trials where the better option was chosen, i.e. proportion of correct choices based on the ground truth reward probability of both cues ($F(3,108)$ = 4.945, $p$ = 0.003, $\eta^2$ = 0.116, Fig 2 and S6 Table). Post-hoc comparisons showed that both SCZ and BPD patients performed significantly worse compared to HC (SCZ–HC $t$ = 3.373, $p_{bonf}$ = 0.006, $d$ = 0.313, BPD–HC $t$ = 3.227, $p_{bonf}$ = 0.01, $d$ = 0.3) across domains (i.e., cue types) (Fig 2 and S6 Table). Overall, response accuracy did not significantly differ between cue types ($F(1,108)$ = 2.577, $p$ = 0.111, $\eta^2$ = 0.019), but

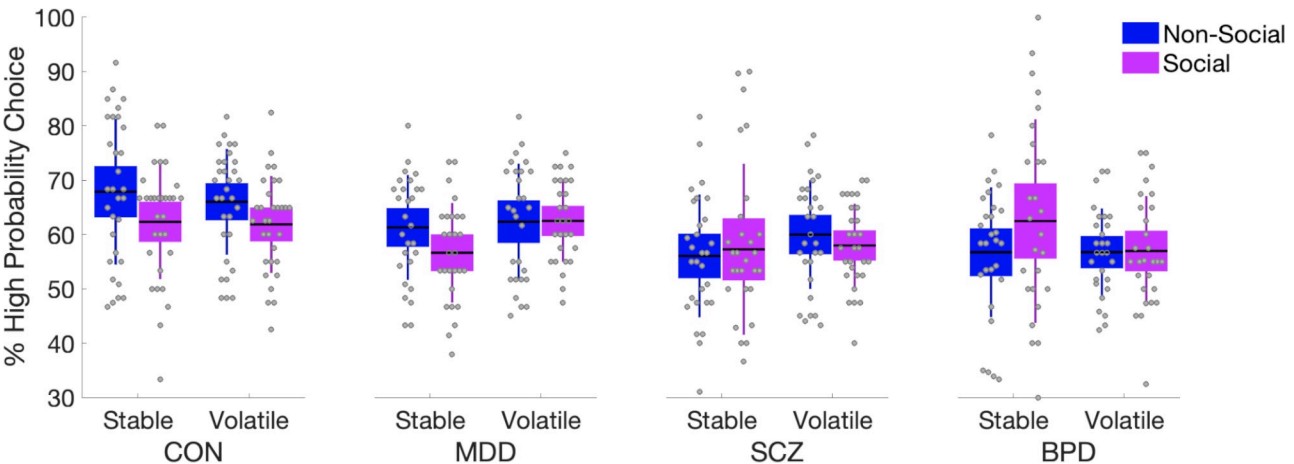

**Fig 2. Proportion of trials where the better option was chosen.** The better option was defined to be the one that according to the ground truth probability schedule was more likely to be rewarded. Importantly, 'better' choice according to color and according to social cue could be different in the same trial. Patients with SCZ and BPD showed poorer response quality in the task compared to HC. A Group x Cue interaction analysis showed that response quality was higher for the non-social cue for HC and MDD patients, whereas BPD patients showed the opposite pattern. Means are plotted with boxes marking 95% confidence intervals and vertical lines showing standard deviations. See also S6 Table.

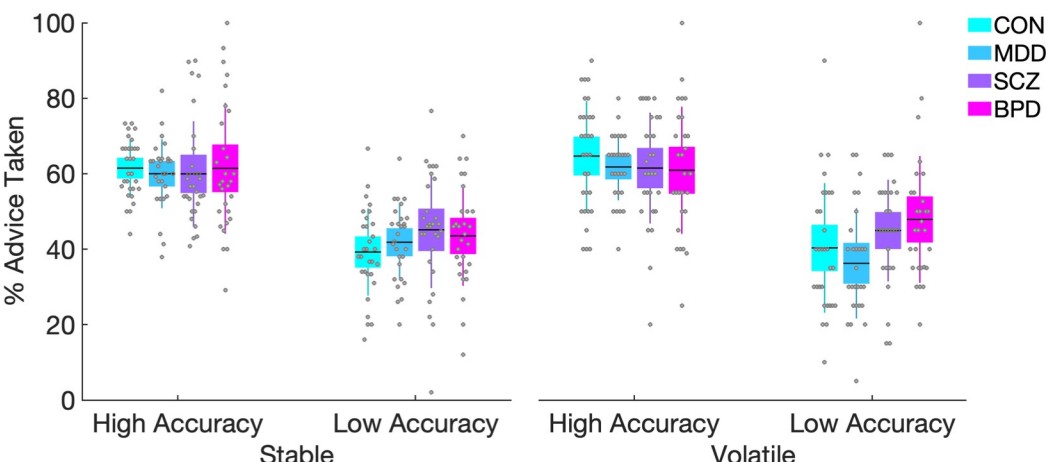

**Fig 3. Behavioral results with regard to advice taking.** All participants followed the advice significantly more during phases of high compared to low accuracy. The descriptive data shows a trend of BPD patients to follow the advice more during volatile phases of low accuracy (right most plot) but the interaction was not significant. Means are plotted with boxes marking 95% confidence intervals and vertical lines showing standard deviations.

there was a significant Cue Type × Group interaction $F(3,108) = 4.820$, $p = 0.003$, $\eta^2 = 0.108$), with HC (Post-hoc: $t(30) = -2.157$, $p_{bonf} = 0.039$, $d = -0.387$) and MDD patients (Post-hoc: $t(27) = -2.181$, $p_{bonf} = 0.038$, $d = -0.412$) showing higher response accuracy with regard to the non-social cue, whereas BPD patients showed the opposite pattern (Post-hoc: $t(27) = 2.058$, $p_{bonf} = 0.049$, $d = 0.389$).

With regard to advice taking behavior during the different phases of the social schedule, we found a main effect of social accuracy ($F(1,108) = 227.935$, $p<0.001$) whereby participants followed the gaze more during phases of high accuracy compared to phases of low accuracy ($t = 14.94$, $p_{bonf} <0.001$) (Fig 3). Advice taking was not significantly affected by the schedule stability ($F(1,108) = 0.503$, $p = 0.480$), indicating that advice taking did not differ between stable and volatile phases. Advice taking was not significantly affected by an interaction between accuracy of the social information and Group ($F(3,108) = 2.222$, $p = 0.09$), or by an interaction between social accuracy, schedule stability and Group ($F(3,108) = 1.47$, $p = 0.227$).

## Bayesian model comparison & validity

Model comparison showed that the HGF including subject specific decision noise as well as the volatility estimate $\hat{\mu}_{3,gaze}$ and $\hat{\mu}_{3,card}$ outperformed the other HGF models, the Rescorla Wagner and Sutton-K1 models with subject specific decision noise only as well as the WSLS and random responding models (PXP = 0.173; XP = 0.914). See Table 1 for further details and

**Table 1. Bayesian model comparison results.**

| BMS | Model 1 | Model 2 | Model 3 | Model 4 | Model 5 | Model 6 | Model 7 | Model 8 |
|---|---|---|---|---|---|---|---|---|
| **EXP_R** | 0.304 | 0.181 | 0.073 | 0.019 | 0.066 | 0.210 | 0.072 | 0.076 |
| **XP** | 0.914 | 0.019 | 0 | 0 | 0 | 0.068 | 0 | 0 |
| **PXP** | 0.173 | 0.119 | 0.117 | 0.117 | 0.117 | 0.122 | 0.117 | 0.117 |

Posterior model probabilities (EXP_R), Exceedance Probabilities (XP) and Protected Exceedance Probabilities (PXP). Model 1 refers to the HGF combined with response model 1, Model 2 refers to the HGF combined with response model 2, Model 3 refers to the HGF combined with response model 3, Model 4 refers to the HGF combined with response model 4, Model 5 refers to the Sutton K-1 Model combined with response model 4, Model 6 refers to the Rescorla Wagner Model combined with response model 4. Model 7 refers to the WSLS Model and Model 8 to the random Model.

S4 Table for mean posterior parameter estimates. We repeated all modeling-based analyses with a reduced sample where all subjects were excluded for whom the random responding model outperformed the others (S4 Table). This did not lead to any qualitative change in results (S7–S9 Tables). We therefore report the results for the full sample here.

### Posterior predictive validity of model parameters and parameter recovery

Subjecting the simulated behavioral readout of % of high probability choices to the same ANOVA as performed with the real behavioral data, we found that the model was able to reproduce the group differences that were observed in the real behavioral data (cf. S6 Table for comparison). The ANOVA of the simulated data showed that for response accuracy, as in the real data, there was a main effect of Group ($F(3,108) = 6.755$, $p = 0.001$, $\eta^2 = 0.149$) with post-hoc t tests revealing significantly lower response accuracy for SCZ and BPD patients compared to HC (SCZ–HC $t = 4.04$, $p_{\text{bonf}} < 0.001$, $d = 0.375$, BPD–HC $t = 3.675$, $p_{\text{bonf}} = 0.002$, $d = 0.341$) and a Group × Cue Type interaction ($F(3,108) = 10.981$, $p < 0.001$, $\eta^2 = 0.219$) showing that HC and MDD patients showed higher response accuracy with regard to the non-social cue, whereas BPD patients showed the opposite pattern (cf. S6 Table and S2 Fig for all results).

Parameter recovery showed that all parameters could be recovered well with the exception of $\omega_{3card}$ (cf. S4 Fig).

### Dynamic learning rates–second level

For the averaged precision weights (i.e., dynamic learning rates) for learning about the social $q(\psi_{2gaze})$ and non-social $q(\psi_{2card})$, we found a main effect of task phase ($F(1,108) = 24.868$, $p < 0.001$, $\eta^2 = 0.182$), showing that $q(\psi_2)$ is higher in volatile compared to the stable phases ($t = -5.147$, $p_{\text{bonf}} < 0.001$, $d = -0.478$) (Fig 4A). In the model-agnostic regression analysis, we observed increased slopes of beta weights over time delays in the volatile vs. stable phase (S5 Fig) indicating higher weighting of most recent trials, which concurs with the increased learning rates in the volatile phase in the HGF. While this effect was observable for both cues, the difference in the model-agnostic analysis was significant only for the card cue, perhaps because this analysis does not consider individual variations in cue use.

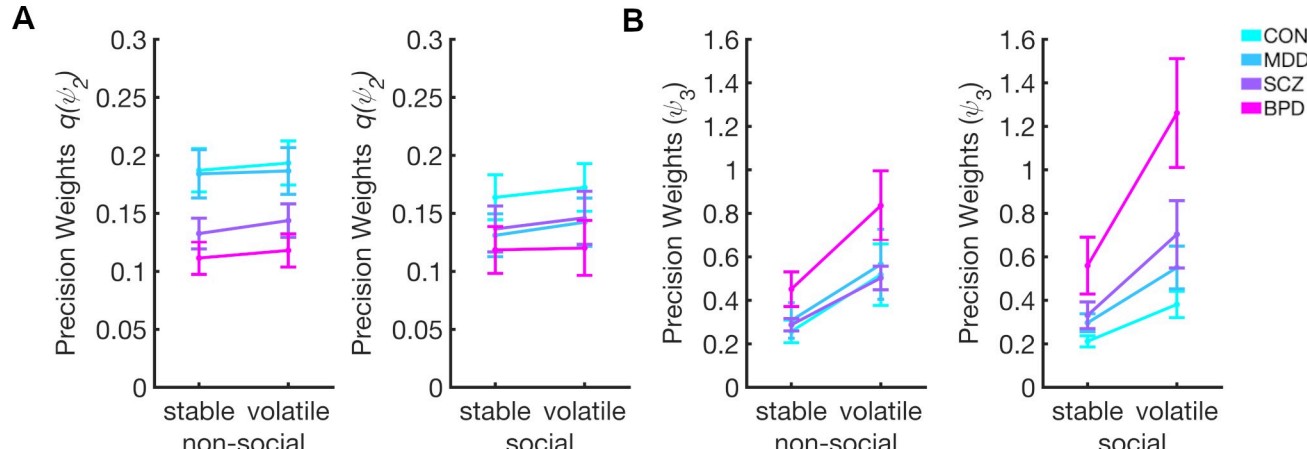

**Fig 4. Results for mixed ANOVA using precision weights for updating beliefs about social and non-social contingency and volatility.** (A) Precision weights $q(\psi_2)$ and (B) precision weights $\psi_3$. Overall, $q(\psi_2)$ and $\psi_3$ increase when transitioning from stable to volatile phase. Patients with BPD show reduced overall $q(\psi_2)$. At same time, patients with BPD show higher $\psi_3$ compared to the other groups and a more pronounced increase in response to volatility. Bars indicate SEM. See also S1 Fig, S7 Table and S8 Table for all results.

There was no significant interaction between Phase and Information Type, which indicates that $q(\psi_2)$ increases similarly during social and non-social volatility ($F(1,108) = 0.131$, $p = 0.718$, $\eta^2 = 0.001$). There was a significant main effect of group ($F(3,108) = 3.557$, $p = 0.017$, $\eta^2 = 0.088$), and the post-hoc $t$-tests revealed that participants with BPD showed significantly lower precision weights on the second level compared to HC ($t = 3.101$, $p_{bonf} = 0.015$, $d = 0.288$). The difference in $q(\psi_2)$ between groups was not affected by Information type ($F(3,108) = 1.038$, $p = 0.379$, $\eta^2 = 0.027$) or its interaction with Phase ($F(3,108) = 0.940$, $p = 0.424$, $\eta^2 = 0.025$) (see S7 Table for all results and for results with the reduced sample).

### Dynamic learning rates–third level

We found a main effect of task phase on precision weights at the third level ($F(1,108) = 116.206$, $p<0.001$, $\eta^2 = 0.462$), showing that $\psi_3$ is higher in volatile compared to the stable phases ($t = -9.784$, $p_{bonf} <0.001$, $d = -0.908$) (Fig 4B). There was a significant main effect of group ($F(3,108) = 6.530$, $p<0.001$, $\eta^2 = 0.141$), and post-hoc t-tests showed that participants with BPD showed significantly higher precision weights at the third level compared to all other groups (BPD–HC $t = -4.204$, $p_{bonf} < .001$, $d = -0.390$; BPD–MDD $t = -3.199$, $p_{bonf} = .011$, $d = -0.297$; BPD–SCZ $t = -3.055$, $p_{bonf} = .017$, $d = -0.284$). In addition, there was a significant Phase × Group interaction ($F(3,108) = 5.962$, $p<0.001$, $\eta^2 = 0.071$) showing that participants with BPD increase their precision weights for both modalities significantly more compared to the other groups when volatility increases. There was a trend of BPD patients showing stronger increases in $\psi_3$ in response to social compared to non social volatility ($F(3,108) = 2.620$, $p = 0.055$, $\eta^2 = 0.065$). The analysis also revealed that $\psi_3$ were affected by the order of schedule ($F(1,108) = 5.008$, $p = 0.027$, $\eta^2 = 0.036$), with $\psi_3$ higher for participants receiving the incongruent-first schedule (i.e gaze starts of being highly misleading) compared to the congruent-first schedule (i.e gaze starts of being highly helpful). This effect was not modulated by Group ($F(3,108) = 2.067$, $p = 0.109$, $\eta^2 = 0.045$) (see S8 Table for all results and for results with the reduced sample).

### Social weighting

The parameter $\zeta$ was a measure of the weight given to the social prediction relative to the learned non-social prediction (cf. Fig 5B & 5C for simulation results). Since $\zeta$ was restricted to the positive domain, estimate distributions were analyzed log-space, where they were less skewed. We found significant group differences in $\log(\zeta)$ ($F(3,108) = 5.893$, $p>0.001$, $\eta^2 = 0.130$ (Fig 5A). Both patients with BPD and patients with SCZ showed significantly higher $\zeta$ estimates compared to controls (BPD: $t = -3.416$, $p_{bonf} = 0.005$; $d = -0.847$, SCZ: $t = -2.855$, $p_{bonf} = 0.031$, $d = -0.691$) but only patients with BPD differed significantly from participants with MDD (BPD: $t = -3.003$, $p_{bonf} = 0.02$, $d = -0.818$; SCZ: $t = -2.451$, $p_{bonf} = 0.095$, $d = -0.650$). Patients with MDD did not show any significant differences compared to controls ($t = -0.335$, $p_{bonf} = 1$, $d = -0.095$). There was a significant main effect of schedule ($F(1,108) = 8.259$, $p = 0.005$, $\eta^2 = 0.061$), showing that participants receiving the congruency-first schedule had higher $\zeta$ compared to participants receiving the incongruency-first schedule ($t = -2.874$, $p_{bonf} = 0.005$, $d = -0.505$). There was no significant interaction between Group and Schedule ($F(3,108) = 0.807$, $p = 0.493$, $\eta^2 = 0.018$). To ensure that the shared mechanism of social over-weighting in BPD and SCZ was not confounded with medication or education status (cf. S1 and S2 Tables), we subjected $\zeta$ to an ANCOVA with $\zeta$ and Chlorpromazine Equivalence Units and Years of School as covariates. The effects were robust to this (cf. S9 Table for results of full and reduced sample).

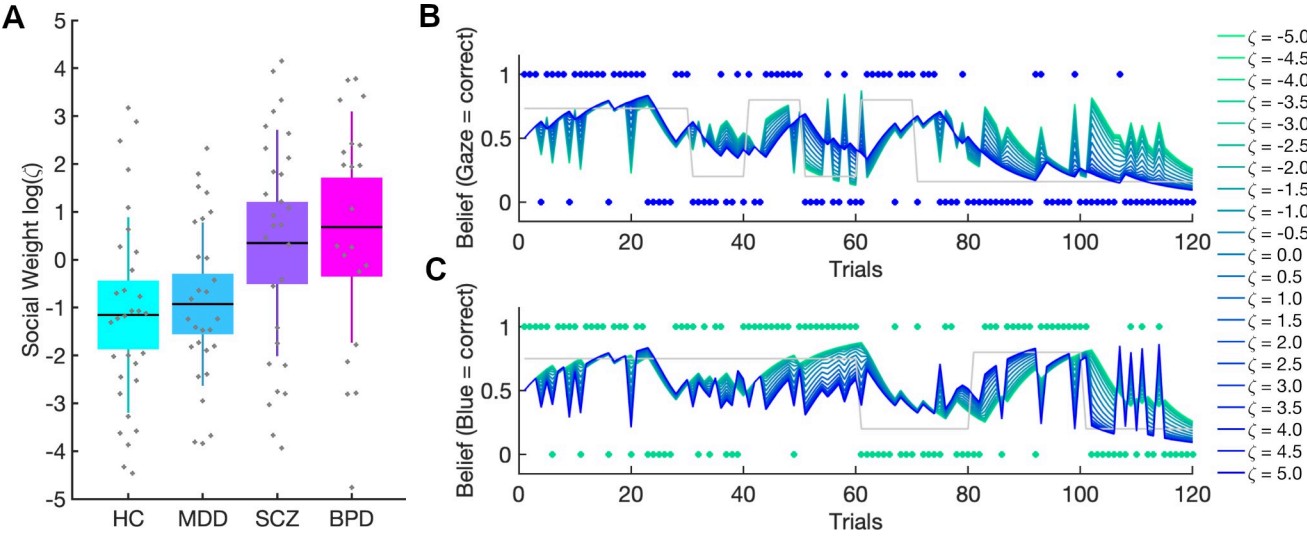

**Fig 5. Social weighting factor log($\zeta$).** (A) Patients with BPD gave the social information significantly more weight compared to HC and patients with MDD. Patients with SCZ also had higher $\zeta$ compared to HC. Boxes mark 95% confidence intervals and vertical lines standard deviations. (B), Simulation results show the impact of varying weighting factor log($\zeta$) on combined belief $b^{(t)}$ (see methods Eq 1). The combined belief $b^{(t)}$ was simulated for agents with same perceptual parameters but different $\zeta$ values (highest values (log($\zeta$) = 5) coded in dark blue, lowest values (log($\zeta$) = -5) in green). (B) shows that the combined belief $b^{(t)}$ of agents with high $\zeta$ values is aligned with the social input structure (blue dots) whereas these agents show a stochastic belief structure with regard to the non-social input structure (green dots) in Panel C. Conversely, agents with low $\zeta$ values show a belief structure closely aligned to the non-social input structure (C), and a stochastic belief structure with regard to the social input (Panel B). The grey lines represent the ground truth of the respective probability schedules. See also S9 Table for all results.

## Social Anhedonia

There was a significant difference in the interpersonal pleasure (ACIPS) ratings between the groups ($F(3,103)$ = 5.719, $p < .001$) (cf. S1 Table). Post hoc $t$ tests revealed that HC showed significantly higher ACIPS scores compared to patients with MDD ($t$ = 3.088, $p_{\text{bonf}}$ = .016) and with BPD ($t$ = 3.802, $p_{\text{bonf}}$ = .001) but not with SCZ ($t$ = 1.833, $p_{\text{bonf}}$ = .418). No significant differences were observed between patients with MDD and SCZ ($t$ = -1.322, $p_{\text{bonf}}$ = 1), patients with MDD and BPD ($t$ = 0.596, $p_{\text{bonf}}$ = 1), nor patients with SCZ and BPD ($t$ = 1.978, $p_{\text{bonf}}$ = 0.304). The multivariate regression using log($\zeta$) and social learning rate $\omega_{2gaze}$ as predictors for ACIPS scores did not show any significant results ($R^2$ = 0.009, $F(2,106)$ = 0.477, $p$ = 0.622).

## Discussion

This study aimed to improve our understanding of the computational mechanisms that underlie the profound interpersonal difficulties in major psychiatric disorders. To achieve this, we used a probabilistic learning task in conjunction with hierarchical Bayesian modeling transdiagnostically in patients with MDD, SCZ, BPD, and healthy controls. The task required participants to perform association learning about non-social contingencies in the presence of a social cue. This allowed us to characterize and quantify the computational aspects of aberrant social inference and decision-making at an individual level. We found that patients with SCZ and BPD showed significantly poorer performance compared to HC and patients with MDD. Patients with MDD performed comparably well to HC. In addition, patients with BPD showed greater response accuracy in the social compared to the non-social domain during the stable phase, while HC and MDD patients showed the opposite pattern. This is particularly remarkable in light of their overall poorer performance. In effect, BPD patients gave up a possible reward advantage by concentrating their learning efforts disproportionately in the social

domain. In addition, we found a tendency in BPD patients to follow the gaze more during volatile phases of low accuracy compared to MDD patients, which however did not reach statistical significance (Fig 3).

These findings raise the question which mechanisms underlie these patterns of behavior. In particular, they call for an investigation of the learning and decision-making mechanisms which give rise to them. Here, computational modeling enabled insights into how beliefs are updated and how these beliefs are translated into decisions: With regard to learning, we found that BPD patients showed increased precision weighting of prediction errors when learning about volatility in both non-social and social information and a tendency for even higher precision weights when learning about social compared to non-social volatility. While volatility learning rates ($\psi_3$) were increased in BPD compared to HC, contingency learning rates ($\psi_2$) were reduced compared to HC both in the social and non-social domain. This accords with a previous finding of blunted social and non-social learning in BPD [19], which was conjectured to result from aberrant volatility beliefs, causing an impairment at detecting contingency changes needed for accurate inference. Because our modeling approach was specifically designed to model beliefs about volatility, it allowed us to address this conjecture. Indeed, our data indicate that impaired contingency learning in BPD is associated with exaggerated learning about environmental volatility. A similar pattern has been observed in autism spectrum disorder (ASD) [8]. Aberrant volatility beliefs in BPD have been suggested to result from unpredictable early-life relationships [19]. However, this is a less likely explanation in ASD which is characterized as a pervasive developmental disorder. This points to a different origin of the mechanistic overlap between our findings and those of [8]. The commonality of aberrant volatility learning may explain the repeated finding of high autism quotient (AQ) values in BPD patients [8], which is confirmed in our sample (cf. S1 Table) and could be taken to suggest a partially shared mechanism of aberrant social inference in these disorders. A previous study from our group found that healthy participants scoring high on AQ showed a similar pattern to the present study's BPD patients, in that they followed the gaze more than low-AQ participants during periods of low accuracy in volatile phases [20]. In that study, computational modeling showed that high AQ participants failed to use the social information to adapt the precision of their belief about the non-social cue. However, this was not found in any participant group in the present study.

This has been the second study demonstrating that aberrant learning in BPD not only concerns social, but also non-social information (cf. [19]). This suggests that aberrant learning occurs independent of domain, in line with previous findings that precision-weighted prediction errors are computed in similar brain regions, irrespective of domain [22,41].

Unlike previous studies on reward [45–47] or volatility [15] learning in SCZ and healthy subjects at risk for psychosis [16], we did not find significant differences between SCZ and HC in this regard. The same applies to MDD patients, where one possible explanation for this negative finding is the lack of punishment for incorrect choices in our task since recent findings converge on impaired aversive learning in depression (e.g. [48,49]).

With regard to decision-making, computational modeling revealed that SCZ and BPD patients both weighted their social-domain predictions more strongly than HC and MDD. This explains the lower performance of BPD and SCZ patients. Their stronger reliance on social cues compared to HC and MDD patients was detrimental because the social cue was more volatile than the non-social one (5 as opposed to 3 contingency changes).

The commonality of over-weighting social-domain predictions in SCZ and BPD patients suggests itself as the decision-making aspect of a general interpersonal hypersensitivity in both conditions [50]. This is also reflected in excessive, albeit inaccurate, mental state attributions (hypermentalizing) [51–53] that constitute a shared feature of BPD and SCZ [6,51–58].

Hypermentalizing is also a possible explanation for the findings of [59], where similar modeling as in the present study showed that healthy participants at the high end of the paranoia spectrum used similar weighting of social information irrespective of whether incorrect advice was framed to be intentional or not, while low-paranoia participants reduced their social weighting when negative advice was cued to be intentional. Furthermore, a study of healthy participants by our group [29] found that stronger weighting of social over non-social predictions during decision-making was associated with increased activity in the putamen and anterior insula. In future studies, it will be interesting to investigate the involvement of these regions in excessive social-weighting and hypermentalizing in BPD and SCZ. Also, a direct comparison of patients with BPD and autism could help to unveil shared mechanisms of aberrant social inference.

In addition to looking for learning and decision-making differences between the different diagnostic groups that were defined by traditional ICD-10 criteria, we adopted a transdiagnostic perspective to investigate the relation between computational mechanisms of social learning and decision-making with ACIPS, a self-report measure of social anhedonia. Previous studies have adopted such a dimensional approach in the general population and found that patterns in aversive learning mapped onto distinct symptoms of depression, social anxiety, and compulsivity [48,59,60]. We, however, could not find any transdiagnostic association of social anhedonia with computational parameters of social learning and decision-making in our data. A possible explanation for this negative finding is that we used scores from a single questionnaire whereas previous studies applied factor analysis (e.g. [60]) on all items from several questionnaires. In addition, these studies investigated larger samples compared to our study (>400 vs. 116) and therefore had more statistical power to detect signficant effects.

## Limitations

We did not use a non-social cue (such as an arrow pointing to a card) as a control condition and therefore cannot fully rule out the possibility that the increased weighting of our social cue observed in BPD and SCZ reflects a more general rather than specifically social peculiarity in information processing. However, eye gaze is a very salient cue and in the paradigm, we aimed to accentuate the social quality of our cue by a clear period of eye contact with the participant before providing the cue.

A further limitation concerns the fact that most patients were in psychopharmacological treatment during data acquisition and had different degrees of disorder severity and chronicity. Furthermore, different patient groups were assessed in different clinical centers, and there was a gender imbalance in the SCZ and BPD groups.

## Conclusion

By adopting a computational psychiatry approach [61–65] to data from an inference task with a social component, we show that BPD patients exhibit an aberrant pattern of learning rate adjustment when the environment becomes more volatile. Instead of quickly relearning changed contingencies, they show exaggerated volatility learning. While SCZ and MDD patients showed a tendency to the same pattern, they did not significantly differ from controls in this respect. We also show that BPD and SCZ patients rely more strongly than controls on social-domain beliefs relative to non-social-domain beliefs when making decisions even in a task where doing the opposite would have given them an advantage. Taken together, this shows that there are computational commonalities as well as differences between patient groups, which suggests some underlying mechanisms that may be shared across diagnoses. Since this approach allows for individually quantifying the severity of impairment at a

mechanistic level, it has the potential to lead to diagnostic and prognostic advances. Furthermore, it points the way to possible targets for novel interventions which transcend traditional diagnostic boundaries.

## Supporting information

**S1 Table. Psychometric data of the participants.** All quantities given as Mean ± SD.
(DOCX)

**S2 Table. Demographic data of the participants.** All quantities given as Mean ± SD.
(DOCX)

**S3 Table. Prior configurations of perceptual and response model parameters.** Means and variances of Gaussian priors are given in the space in which the parameter was estimated (native, log, or logit).
(DOCX)

**S4 Table. Within-subjects model comparison.**
(DOCX)

**S5 Table. Mean posterior estimates of learning model and decision model parameters estimated from winning model.**
(DOCX)

**S6 Table. Statistics for mixed ANOVA with response accuracy (% High probability choices) from real and simulated data for stable and volatile phases (Factor Phase) of social and non-social cue (Factor Cue Type) for all groups (Factor Group) and schedules (Factor Schedule).** The table shows the results for the real and simulated behavior.
(DOCX)

**S7 Table. Statistics for mixed ANOVA with averaged $q(\psi_2)$ during stable and volatile phases (Factor Phase) of social and non-social cue (Factor Cue Type) for all groups (Factor Group) and schedules (Factor Schedule).** The table shows the results for the full and reduced sample.
(DOCX)

**S8 Table. Statistics for mixed ANOVA with averaged $\psi_3$ during stable and volatile phases (Factor Phase) of social and non-social cue (Factor Cue Type) for all groups (Factor Group) and schedules (Factor Schedule).** The table shows the results for the full and reduced sample.
(DOCX)

**S9 Table. Statistics for ANCOVA with $\zeta$ and Chlorpromazine Equivalence Units and Years of School as covariate for full and reduced sample.**
(DOCX)

**S1 Fig. Learning trajectories for one example participant.** A, Precisions $\psi_{3card}$ (red) and $\psi_{3gaze}$ (blue) that modulate the weight on B, prediction errors $\delta_{2card}$ (red) and $\delta_{2gaze}$ (blue). C, Precision weights $\psi_{2card}$ in red trajectory and $q(\psi_{2card})$ in red dotted trajectory. Precision weights $\psi_{2gaze}$ in blue trajectory and $q(\psi_{2gaze})$ in blue dotted trajectory. Precision weights modulate weight on D) prediction error $\delta_{1card}$ (red) and $\delta_{1gaze}$ (blue) signals. E, Dark red dots mark the input structure of the non-social information (blue correct = 1; green correct = 0) and the dotted red line represents the ground truth of this input structure. Light red dots mark the choices (blue card = 1; green card = 0). The red trajectory is the participant specific belief

trajectory about the blue card to be correct that was estimated on the basis of the choices. E, The same logic applies to the social input and response structure in blue. The posterior parameter estimates for this particular participant were $\omega_{2card}$ = -1.460, $\omega_{2gaze}$ = -3.576, $\omega_{3card}$ = -6.021, $\omega_{2gaze}$ = -6.074, $\log(\zeta)$ = -2.0243, $\log(\beta)$ = 2.207, $\text{logit}(\eta)$ = 0.211.
(TIF)

**S2 Fig. Posterior Predictive Validity.** Simulated Behavior from the posterior estimates of all participants revealed the same effects as real behavior. Boxes mark 95% confidence intervals and vertical lines standard deviations.
(TIF)

**S3 Fig. Grouped individual data points showing precision weights for updating beliefs about social and non-social contingency and volatility.** A, precision weights $q(\psi_2)$. B, precision weights $\psi_3$. Overall, $q(\psi_2)$ and $\psi_3$ increase when transitioning from stable to volatile phase.
(TIF)

**S4 Fig. Parameter recovery.** We simulated behavioral responses based on the posterior estimates of all participants 10 times, resulting in 1160 simulations. For each subject, we calculated the average posteriors estimated from the simulated data (x-axis) and correlated (Pearson's correlations) them with the original posterior parameters estimates (y-axis), which is shown in plots a-f. The estimated parameters could be recovered well, however, the third level evolution rate ($\omega_{card}$) could not be recoverd well.
(TIF)

**S5 Fig. Regression model of participants choices for both cue types.** Top: Regression model for choices in card space (1 = taking blue, 0 = taking green) with predictors of card accuracy (1 = blue correct, 0 = green correct) for the past 5 trials, and reward values (reward value if blue taken, reward value if green taken) for the whole task (left), stable (middle) and volatile (right) phase. The slope between predictors were calculated as a model-agnostic readout of the 'learning rate', indiciating the degree to which more recent information is weighted. A t-test was applied to compare the difference between these slopes during stable and volatile phases of the task. Slopes are increasing during volatile compared to stable phases. Below: Regression model for choices in gaze space (1 = taking advice, 0 = not taking advice) with predictors of gaze accuracy (1 = gaze correct, 0 = gaze incorrect) for the past 5 trials, and card reward values (reward value if advice taken) for the whole task (left), stable (middle) and volatile (right) phase.
(TIF)

## Acknowledgments

Lara Henco would like to thank the Graduate School of Systemic Neurosciences (LMU). Dr. Andreea Diaconescu was supported by the Swiss National Foundation (PZ00P3_167952) and the Krembil Foundation. We would like to thank Lea Duerr, Nina von Aken, Mathilda Sommer and Benjamin Pross for helping with the recruitment and data collection.

## Author Contributions

**Conceptualization:** Lara Henco, Andreea O. Diaconescu, Leonhard Schilbach.

**Data curation:** Lara Henco.

**Formal analysis:** Lara Henco, Andreea O. Diaconescu, Juha M. Lahnakoski, Christoph Mathys.

**Funding acquisition:** Leonhard Schilbach.

**Investigation:** Lara Henco, Marie-Luise Brandi, Sophia Hörmann, Johannes Hennings, Alkomiet Hasan, Irina Papazova, Wolfgang Strube.

**Methodology:** Lara Henco, Andreea O. Diaconescu, Leonhard Schilbach, Christoph Mathys.

**Project administration:** Lara Henco, Leonhard Schilbach.

**Resources:** Johannes Hennings, Alkomiet Hasan, Leonhard Schilbach, Christoph Mathys.

**Software:** Andreea O. Diaconescu, Juha M. Lahnakoski, Christoph Mathys.

**Supervision:** Leonhard Schilbach, Christoph Mathys.

**Validation:** Lara Henco, Andreea O. Diaconescu, Dimitris Bolis, Christoph Mathys.

**Visualization:** Lara Henco.

**Writing – original draft:** Lara Henco.

**Writing – review & editing:** Lara Henco, Andreea O. Diaconescu, Juha M. Lahnakoski, Marie-Luise Brandi, Sophia Hörmann, Johannes Hennings, Alkomiet Hasan, Irina Papazova, Wolfgang Strube, Dimitris Bolis, Leonhard Schilbach, Christoph Mathys.

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
