## [Decision Letter · Decision Letter 0]

24 Apr 2020

Dear Ms Henco,

Thank you very much for submitting your manuscript "Aberrant computational mechanisms of social learning and decision-making in schizophrenia and borderline personality disorder" for consideration at PLOS Computational Biology.

As with all papers reviewed by the journal, your manuscript was reviewed by members of the editorial board and by several independent reviewers. In light of the reviews (below this email), we would like to invite the resubmission of a significantly-revised version that takes into account the reviewers' comments.

As you will see, the reviewers were generally positive about your manuscript. However, they had a few serious concerns that need to be addressed. In particular, we would like to call your attention to reviewer 2's comment regarding the comparability of social vs. non-social learning. Even though this point was already mentioned under Limitations, any conclusions regarding social vs. non-social learning should probably made with more caution. Another point, which was raised by several reviewers, is that demonstration of parameter recovery and model simulations would considerably strengthen the paper. We cannot make any decision about publication until we have seen the revised manuscript and your response to the reviewers' comments. Your revised manuscript is also likely to be sent to reviewers for further evaluation.

Sincerely,

Philipp Sterzer

Guest Editor

PLOS Computational Biology

Samuel Gershman

Deputy Editor

PLOS Computational Biology

Reviewer's Responses to Questions

**Comments to the Authors:**

Reviewer #1: In this paper, Henco et al. evaluate alterations in hierarchical evidence in social and non-social contexts across a transdiagnostic sample comprising individuals with schizophrenia, major depressive disorder, borderline personality disorder and healthy controls. They use a well-designed advice taking paradigm that manipulates of volatility and model the data using a variant of the 3-level hierarchical Gaussian filter, a widely used approach developed by this group. The experimental design and computational modeling is rigorous and sound and the transdiagnostic evaluation is a relevant and novel aspect of this work.

While this work has some relevant limitations, the strengths seem to outweigh them. I am generally enthusiastic about this work and only have relatively minor comments that I would like the authors to address.

1. One potential concern has to do with low-level confounds related to poor accuracy or use of heuristics. On Fig 3a, it seems like response accuracy is around chance level in some subjects. Can the authors provide evidence that these subjects understand task instructions and engage with the task as intended? Any basic manipulation checks would be useful. Furthermore, can the authors show that the group differences in fitted model parameters are not driven by low-level issues such as low overall poor accuracy or use of heuristics? Looking for strategies such as win-stay/lose-shift and the percentage of participants using them in each sample would also be advisable. Perhaps a within-subject model comparison could help exclude subjects for whom there is no evidence to support they are learning during the task.

2. One important aspect of this study is the transdiagnostic evaluation of social learning impairments in relation to the ACIPS measure of social anhedonia, which the authors report as a negative finding. This result should be further emphasized in the discussion for balance. Previous work (e.g., Gillan et al.) suggests that transdiagnostic measures are more related to computational phenotypes than DSM categories, but this work seems to suggest otherwise. While this is a nuanced point, the authors should discuss the implications of this negative result along the lines of relevant previous work.

3. The authors make an effort to explain the partially convergent findings in schizophrenia and borderline personality disorders in terms of their shared symptoms, a relevant point that enriches the discussion. Obviously, these disorders also have many aspects that are not shared. Can the authors conduct further analyses to support which specific shared symptom domains may be driving the convergent findings across these two disorders? Can these findings be instead due to other confounds that may be shared between these two disorders or the specific samples included here, for instance similar medication or lower IQ or differences in sociodemographic status?

4. Related to the previous point, one aspect I would like the authors to emphasize further is the integration of the current results with their previous results in autism and paranoia in the general population.

5. It is unclear why the authors decided to simulate the data for a subset of participants in multiple iterations. Can’t the authors simulate data using the fitted parameter for each subject and take average responses for several iterations in a given subject as the responses in that subject so as to avoid differences in sample size and statistical power between the real and the simulated data? This needs clarification.

6. I would like for the authors add a parameter recovery analysis supporting the identifiability of parameters in their winning model, which tends to be a concern with highly complex models such as the one described here.

Minor points:

7. In Fig. 3 it would be preferable to show scattered data points for each subject (at least for the stable versus volatile contrast estimates).

8. Typo on line 149: I believe they mean ‘afforded’ instead of ‘accorded’

9. To clarify, I am assuming these are all new samples, or have any of them been previously reported on? If so, please indicate which subsamples or parts of samples have been included in previous manuscripts.

Reviewer #2: This study aims at assessing probability learning and volatility in schizophrenia (SCZ), borderline personality disorder (BPD) and major depressive disorder (MDD) as well as learning from social and non social cues.

The authors test patients diagnosed with MDD (N=29), SCZ (N=31), and BPD (N=31), and healthy controls (N=34) performing a probabilistic reward learning task in which participants could learn from social (a cue given by the gaze of a person in a photo) and nonsocial information regarding probabilities of two cards (a blue card and a green card) to give a reward. The probabilities of the pairing between each card and rewards could be stable or volatile. The social cue could be congruent or incongruent with the correct answer.

The individual behaviour data was fit using a HGF model.

Participant also filled an AQ questionnaire and a questionnaire regarding social anhedonia.

The raw data showed that BPD and SCZ had worse performance than MDD and HC with patients with BPD following the social advice significantly more compared to patients with MDD during volatile phases of low accuracy.

The modelling suggests impaired learning from social and non-social information in BPD characterised by an exaggerated sensitivity to changes in environmental volatility. Compared to controls, patients with BPD and SCZ showed an over-reliance on their beliefs about the predictive value of social relative to non-social information during decision-making.

The topic is very timely and interesting. The paper is clear and well written (though fairly technical due to being centred on the use of the HGF model). I think using this sort of task and modelling with patients having different types of ICD10 diagnostics and looking at commonalities or differences is very interesting and what needs to be done to progress in the field.

However, I have 2 major concerns about the study:

1) As the authors point out in Discussion, although the authors want to assess “social learning” (as per the title of the article), they didn’t use a control condition to assess whether the differences observed with the “social cue” had anything to do with the “social” nature of the cue (i.e. that it is presented as an eye-gaze), or whether it is more related to having to integrate two types of cues, that is whether they are looking at a difference in cue integration rather than in social learning, where the two cues might also rely on working memory in different ways.  They could have used a control condition with an arrow for example instead of a gaze, and I don’t really understand why the authors didn’t do that.  In my mind, it would have greatly enhanced the study and justified framing it as a study about “social learning”. In the absence of such a control condition, I find that all the arguments related to social learning to be weak and speculative.   

2) I think because the study relies on the complex HGF model, it would require showing how good the model is at parameter and model recovery for this task/data (cf. Wilson and Collins, Ten simple rules for the computational modelling of behavioural data, eLife 2019).  The model comparison with simpler models in particular the Rescorla-Wagner model (with fixed learning rate) doesn’t sound very fair: how does the RW model include the social cue?

 It is important to justify the use of such complex models (rather than starting from them), and in my mind the way to do this is both to show that parameter and model recovery works, and that no simpler model can account for the data.

Minor comments:

- Table 1 - explain how the reader is supposed to make sense of those numbers

- Abstract and main text could be clearer in terms of what the behaviour shows, what questions it raises and what the modelling reveals in addition.

Reviewer #3: The present study applied a probabilistic reinforcement learning task to three different groups of patients, major depressive disorder (MDD), schizophrenia (SZ), and borderline personality disorder (BPD), and healthy controls. The task allowed directly learning from the outcome of choices and about the reliability of a social cue (the gaze of a face presented centrally between the two choice options). Hierarchical Gaussian filter, a variant of Bayesian inference was applied to the behavioural data to extract different learning parameters. The main finding is that BPD show diminished learning from either source of information (outcomes and gaze information), whilst at the same displaying aberrantly high learning about the volatility of the environment (both the outcome and gaze, which orthogonally varied). Furthermore, choices of both BPD and SZ were guided more strongly by gaze compared to outcome information relative to healthy controls.

Overall, this is a very interesting study and I really appreciate the attempt at linking different clinical diagnoses with underlying computational phenotypes. However, at present, I am not fully confident about the findings presented. Specifically:

1) The group differences are exclusively observed in the model-derived parameters, not in any of the behavioural parameters. It is therefore very important to be able to follow exactly on what the HGF is doing. At present, I find the description of the HGF not very intuitive to follow for readers not familiar with Bayesian inference. Thus, one would have to take the authors' word for it. Rewriting this paragraph in a way that makes it more amenable to non-modellers could provide a lot more clarity - and make the work of greater interest to a broader readership.

2) Relatedly, I would find it very assuring if the authors could present a behavioural readout that does not rely on high-level Bayesian modelling. One option that comes to mind is to use a regression-based approach. Entering the (blue) outcomes and the gaze direction, each for the past n trials, reward available on the current trial (and possibly the interaction of both) could be used as predictors. This could be run separately for the stationary and volatile environments. In my view, the slope of the (social and non-social) weights should reflect the social and non-social learning rates, and the difference between blocks would reflect their adjustments in response to changes in volatility.

3) It is currently hard to judge how well the best fitting model described participants' choices. Could you please provide some more intuitive measure, such as the model's average probability of choosing the option selected by the participant? Is the winning model the same in all patient groups/controls? In how many of the participants was it the best one?

4) Only two non-Bayesian models are presented, and these are not explained, the reader is referred to two references. While I know what an RW model is, I had not heard about the Sutton K1 model before, and even a simple RW model can come in many flavours. What was it that motivated the choice of these specific two non-Bayesian models? Why not others, e.g. some that also feature dynamic learning rates (e.g Pearce-Hall associability), or separate learning rates for social and non-social information? There are various other alternative models that could be compared here, and I am not sure why the authors restricted it to those two.

5) I am also missing details on the task. What was the range of possible outcomes? I am asking because (as described by prospect theory) humans typically do not weight reward amount linearly, tending to underweight higher amounts. Does it improve model fits to account for such subject-specific distortions?

6) In the results, it says: "We simulated responses using the posterior mean parameter values of 60 randomly chosen participants from the best fitting model to demonstrate that this model was capable of reproducing the group differences that were observed in the real behavioral data..."

This seems arbitrary. Why only a subset (of more than 50% of all participants) rather than all participants? Furthermore, can the fitted parameters be recovered from the simulated data?

7) Overall performance (% correct) is impaired in BDP and SCZ, and overall task performance is quite low. In particular in the patient group, there appears to be a significant fraction of people that performs just around chance level (some slightly above, some even below). This leaves me wondering whether there was a substantial proportion of participants (patients) that failed to properly perform the task?

Minor:

1) How did you define "% correct"? Does this refer to choices of the option with higher probability or with higher expected value (probability x points)?

2) Figure 2: please indicate in the legend what values are shown. It looks like medians in 2A+B, but mean ± SEM in 2C? Please clarifiy.

3) Parts of the abstract appear to be redundant, unless I'm mistaken, lines 72-76 merely repeat the findings already stated above?

**Have all data underlying the figures and results presented in the manuscript been provided?**

Reviewer #1: Yes

Reviewer #2: No: At the moment they state "All XXX files are available from the XXX database (accession number(s) XXX, XXX.)." (haven't replaced the XXX).

Reviewer #3: None

PLOS authors have the option to publish the peer review history of their article (what does this mean?). If published, this will include your full peer review and any attached files.

Reviewer #1: No

Reviewer #2: No

Reviewer #3: No
---

## [Decision Letter · Decision Letter 1]

19 Jul 2020

Dear Ms. Henco,

We are pleased to inform you that your manuscript 'Aberrant computational mechanisms of social learning and decision-making in schizophrenia and borderline personality disorder' has been provisionally accepted for publication in PLOS Computational Biology.

Best regards,

Philipp Sterzer

Guest Editor

PLOS Computational Biology

Samuel Gershman

Deputy Editor

PLOS Computational Biology

Reviewer's Responses to Questions

**Comments to the Authors:**

Reviewer #1: The authors have been very responsive and adequately addressed all my comments.

Reviewer #2: The authors have answered a lot of my concerns. I am still not completely convinced about the lack of a control condition to test whether the social nature of the cue is really the determining factor in how it is integrated. However, the authors have now expanded their discussion of this aspect. Overall, I think the paper is now greatly improved and will be very interesting to the community.

Reviewer #3: Thank you for the thorough revisions and the detailed additional analyses. One of my concerns had been that a significant fraction of participants displayed random responding. One of the new analyses shows that a model with random responding indeed is the winning model in n = 16 volunteers. Importantly, however, their main results appear immune to this (excluding these volunteers does not change the main pattern of results) and the state of things is clearly and transparently presented to the reader.

The model-agnostic analyses, in my view, provide important support for the modeling-based claims.

Overall, I think the manuscript now is in good shape for publication.

**Have all data underlying the figures and results presented in the manuscript been provided?**

Reviewer #1: Yes

Reviewer #2: Yes

Reviewer #3: None

PLOS authors have the option to publish the peer review history of their article (what does this mean?). If published, this will include your full peer review and any attached files.

Reviewer #1: **Yes: **Guillermo Horga

Reviewer #2: No

Reviewer #3: No

---

## [Editor Report · Acceptance letter]

24 Sep 2020

PCOMPBIOL-D-20-00401R1 

Aberrant computational mechanisms of social learning and decision-making in schizophrenia and borderline personality disorder

Dear Dr Henco,

I am pleased to inform you that your manuscript has been formally accepted for publication in PLOS Computational Biology. Your manuscript is now with our production department and you will be notified of the publication date in due course.

With kind regards,

Matt Lyles
